# Entelon^®^ (*Vitis vinifera* Seed Extract) Prevents Cancer Metastasis via the Downregulation of Interleukin-1 Alpha in Triple-Negative Breast Cancer Cells

**DOI:** 10.3390/molecules26123644

**Published:** 2021-06-15

**Authors:** Daeun You, Yisun Jeong, Sun Young Yoon, Sung A Kim, Eunji Lo, Seok Won Kim, Jeong Eon Lee, Seok Jin Nam, Sangmin Kim

**Affiliations:** 1Department of Breast Cancer Center, Samsung Medical Center, 81 Irwon-ro, Gangnam-gu, Seoul 06351, Korea; daeun0815@naver.com (D.Y.); seokwon1.kim@samsung.com (S.W.K.); paojlus@hanmail.net (J.E.L.); 2Department of Health Sciences and Technology, SAIHST, Sungkyunkwan University, 81 Irwon-ro, Gangnam-gu, Seoul 06351, Korea; sunrise1526@naver.com (Y.J.); soso66772000@naver.com (S.Y.Y.); tjddk5920@naver.com (S.A.K.); rej4178@naver.com (E.L.); 3Department of Surgery, Samsung Medical Center, Sungkyunkwan University School of Medicine, 81 Irwon-ro, Gangnam-gu, Seoul 06351, Korea

**Keywords:** Entelon, IL-1, lung metastasis, triple-negative breast cancer

## Abstract

Interleukin-1 (IL1) is a proinflammatory cytokine and promotes cancer cell proliferation and invasiveness in a diversity of cancers, such as breast and colon cancer. Here, we focused on the pharmacological effect of Entelon^®^ (ETL) on the tumorigenesis of triple-negative breast cancer (TNBC) cells by IL1-alpha (IL1A). IL1A enhanced the cell growth and invasiveness of TNBC cells. We observed that abnormal IL1A induction is related with the poor prognosis of TNBC patients. IL1A also increased a variety of chemokines such as CCL2 and IL8. Interestingly, IL1A expression was reduced by the ETL treatment. Here, we found that ETL significantly decreased the MEK/ERK signaling pathway in TNBC cells. IL1A expression was reduced by UO126. Lastly, we studied the effect of ETL on the metastatic potential of TNBC cells. Our results showed that ETL significantly reduced the lung metastasis of TNBC cells. Our results showed that IL1A expression was regulated by the MEK/ERK- and PI3K/AKT-dependent pathway. Taken together, ETL inhibited the MEK/ERK and PI3K/AKT signaling pathway and suppressing the lung metastasis of TNBC cells through downregulation of IL1A. Therefore, we propose the possibility of ETL as an effective adjuvant for treating TNBC.

## 1. Introduction

Breast cancer is the most common cancer in women around the world, and can be classified into four molecular subtypes: luminal A, luminal B, human epidermal growth factor receptor 2 (HER2)-enriched, and basal-like types by genetic profiling or by immunohistochemical (IHC) staining techniques [1,2,3]. Basal-like breast cancer is commonly known as triple-negative breast cancer (TNBC), comprising ~15% of all breast cancers. [4]. TNBC is more biologically aggressive than other subtypes, characterized by a higher risk of recurrence within five years of diagnosis [5,6,7]. Unfortunately, TNBC is a heterogeneous disease without clinically approved therapeutic target drugs against interleukin-1 (IL-1).

Interleukin-1 (IL-1) cytokines are major inflammatory cytokines and are upregulated in several types of cancers, including breast, lung, and melanoma [8,9]. Induction of IL-1 promotes angiogenesis, tumor growth, and metastasis in malignant melanoma and breast cancer in vivo [10]. IL-1 also enhances the synthesis of various chemokines or cytokines, such as IL6, IL8, chemokine (C-C motif) ligand 20 (CCL20), and vascular endothelial growth factor (VEGF) [11,12]. Moreover, patients generating high levels of IL-1 have a poor prognosis [13,14]. IL-1 is an attractive therapeutic target, and its inhibitors are applied to treat a wide range of malignancies [15,16].

Grape seed extract (GSE, Entelon^®^ (ETL)), a widely used dietary supplement, is known to have various pharmacological effects, including antioxidant, anti-inflammatory, anti-tumor, and anti-aging activity [17]. GSE contains a high content of flavonoid polyphenolic compounds including gallic acid, (+)-catechins, (−)-epicatechins, ferulic acid, and proanthocyanidin [17,18]. GSE is also linked to cancer prevention and the increased consumption of grapes is associated with reduced cancer risk [19]. GSE prevents angiogenesis by suppressing the VEGF/VEGF receptor (VEGFR) signaling pathway and the upregulation of insulin-like growth factor binding protein-3 [20,21]. Currently, ETL is used in South Korea as an adjuvant treatment for lymphadenoma due to breast cancer treatment or to improve symptoms related to venous lymph node failure.

Here, we mainly focused on the inhibitory effects of ETL on the tumorigenesis of TNBC cells. First, we studied how IL1A controls the tumor growth and invasion of TNBC cells and how ETL regulates the level of IL1A expression. These results provide the potential of ETL for IL1A targeting therapies for TNBC.

## 2. Results

### 2.1. ETL Downregulates Mouse IL1A Suppression in 4T1 TNBC Cells

To confirm the pharmacological activity of ETL, we treated 4T1 TNBC cells with ETL at the indicated doses for 48 h. As shown in Figure 1A, cell viability was reduced to 70% by 60 μg/mL ETL treatment. Then, we conducted an experiment using 30 μg/mL ETL, since we confirmed that ETL did not significantly affect cell viability at concentrations below 60 μg/mL ETL. Furthermore, we also checked the effect of ETL on the cell cycle. At 30 μg/mL ETL, the cell growth and cell cycle ratio was not significantly changed by ETL (Figure 1B,C). Next, we investigated the effect of ETL on the expression of secreted cytokines using the Proteome Profiler Mouse Cytokine Array. Interestingly, the IL1A expression level was decreased by ETL treatment (Figure 1D). Under similar conditions, we measured the levels of IL1A mRNA expression in 4T1 TNBC cells. As shown in Figure 1E, the IL1A mRNA expression level was dose-dependently decreased by ETL. The level of IL1A mRNA expression was reduced by 0.47 ± 0.02-fold compared to the controls by treatment with 30 μg/mL ETL (Figure 1E).

### 2.2. mIL1A Augments Cell Growth and Invasiveness in TNBC Cells

We investigated the role of mIL1A in the tumorigenicity of 4T1 TNBC cells. To demonstrate the role of mIL1A in the growth of 4T1 TNBC cells, we treated cells with or without 20 ng/mL mIL1Afor two weeks. As shown in Figure 2A, mIL1A treatment increased cell growth 177.3 ± 22.8-fold compared to the controls. In addition, we also examined the effect of mIL1A on cell invasiveness. As expected, mIL1A significantly increased the invasiveness of 4T1 TNBC cells compared to the controls (Figure 2B). Next, we studied the effect of mIL1A on tumorigenesis of TNBC cells in vivo. The schematic model of the experimental procedure is shown in Figure 2C. 4T1 TNBC cells were injected with or without 20 ng/mL mIL1A into the second fat pad of the mice. We measured the tumor size during 15 days. As shown in Figure 2D, tumor size was increased by treatment with mIL1A. Therefore, we demonstrated that mIL1A enhanced tumorigenesis including tumor growth and invasiveness in 4T1 TNBC models.

### 2.3. IL1A Induction Is Related to Recurrence Rates in TNBC Patients

To explore the clinical significance of IL1A in TNBC patients, we evaluated whether the level of IL1A could represent a prognostic biomarker for TNBC patients. Thus, we analyzed DNA microarray-based gene expression data using the KM plotter database [22]. Here, we analyzed the prognostic value of IL1A mRNA abundance in TNBC patients. As shown in Figure 3, our results showed that TNBC patients with a high level of IL1A had a poor prognosis. Breast cancer patients with a high expression of IL1A showed poorer RFS (*p =* 0.042, Figure 3A) and DMFS (*p =* 0.037, Figure 3B) than the patients with low expression.

Next, we investigated the effect of IL1A on the invasiveness of MDA231 and Hs578T TNBC cells using a Matrigel-coated Boyden chamber assay. As shown in Figure 4A, IL1A-treated MDA231 and Hs578T TNBC cells showed significantly higher invasive activity than that of the control cells. The number of invading cells was increased to 435.3 ± 32.8% (in MDA231 cells) and 260.8 ± 19.5% (in Hs578T cells) by 20 ng/mL IL1A treatment (Figure 4A). Changes in the inflammatory chemokines by IL1A were also analyzed. The level of CCL2 and IL8 mRNA expression in TNBC cells was increased by IL1A treatment (Figure 4B,C, and Supplement 1). Thus, TNBC patients with high IL1A were associated with poor prognosis. The secretion of IL1A induced cell invasiveness and inflammatory chemokines in TNBC cells.

### 2.4. ETL Downregulates Human IL1A Expression in MDA231 TNBC Cells

We examined the effect of ETL on IL1A expression in human TNBC cells. As shown in Figure 5A, ETL did not significantly affect cell growth at the indicated concentrations. Based on these ETL concentrations, we treated MDA231 TNBC cells with 15 or 30 μg/mL ETL for 24 h. The level of IL1A mRNA expression, but not IL1B expression was reduced by ETL treatment (Figure 5B). Under similar conditions, the level of secreted IL1A protein was also reduced by ETL (Figure 5C). The concentration of secreted IL1A protein in MDA231 TNBC cells was reduced by 1.70 ± 0.51 pg/mL of control level (29.90 ± 0.51 pg/mL) by 30 μg/mL ETL treatment (Figure 5C).

### 2.5. ETL Downregulates Human IL1A Level by Preventing the MEK/ERK and PI3K/AKT Signaling Pathway in MDA231 TNBC Cells

To explore the regulatory mechanism of IL1A expression, we treated MDA231 TNBC cells with various signaling molecule inhibitors for 24 h. The level of IL1A mRNA was reduced by UO126 or LY294002, but not by SP600125 or Stattic (Figure 6A). Under similar conditions, the concentration of secreted IL1A protein was reduced by UO126 or LY294002 treatment (Figure 6B). The level of secreted IL1A protein was reduced by 2.21 ± 1.29 pg/mL and 10.29 ± 0.14 pg/mL of the control level (55.66 ± 1.93 pg/mL) by the treatment of MDA231 TNBC cells with 5 μM UO126 or LY294002, respectively (Figure 6B). Next, we examined the suppressive effect of ETL on various signaling pathways. The cells were treated with the indicated concentrations for 24 h. The levels of phosphorylated ERK and AKT were decreased by ETL (Figure 6C). Our results indicate that ETL inhibited IL1A expression through the suppression of the MEK/ERK- and PI3K/AKT-dependent pathway in TNBC cells.

### 2.6. ETL Prevents the Tumor Growth and Metastatic Capacity of TNBC Cells

Lastly, we studied the pharmacological effect of ETL on tumor growth and metastasis using an orthotopic xenograft model. The schematic model of the experimental procedure is shown in Figure 7A. Increases in the tumor volume were delayed by 0.2% ETL treatment (Figure 7B). We also analyzed the expression of H&E, Ki67, and TUNEL in tumor tissues from each condition (Figure 7C). Although the expression of Ki67 was slightly reduced by ETL, there was no significant difference (Figure 7C). However, our results showed that the TUNEL-positive cells, which are markers of apoptosis, were increased in 0.2% ELT treated group compared to vehicle-only-treated group (Figure 7C).

Next, we studied the suppressive effect of ETL on metastasis using 4T1 mammary carcinoma cells, which are highly tumorigenic and invasive model. We injected 4T1 TNBC cells into the second fat pad of the mice. The metastatic capacity of 4T1 TNBC cell orthotopic xenograft tumors was prevented by ETL treatment (Figure 7D). The number of lung metastatic nodules was reduced in ETL-treated group (13.52 ± 1.87 nodules) compared to vehicle-treated group (44.83 ± 8.43 nodules) (Figure 7D). We also performed histological analysis of the lungs using H&E staining. In ETL-treated group, the number of metastatic sites was dramatically reduced in ETL-treated group compared to vehicle-treated group (Figure 7D). Thus, our results indicate that ETL suppressed tumor growth and metastasis of TNBC cells.

## 3. Discussion

GSE is known to have many beneficial health properties, including protection against hypertension [23], the prevention and treatment of diabetes [24], and the prevention of obesity [25]. In addition, the incidence and growth of UVB radiation-induced photocarcinogenesis of skin cancer was effectively prevented by the dietary feeding of GSE [26]. To enhance the efficacy of GSE, Sharma et al. reported that the combination of the cytotoxic agent doxorubicin and GSE exerted a synergistic effect on inhibiting the growth of breast cancer cells [27]. As mentioned in the introduction, GSE (Entelon^®^ (ETL) supplied by Hanlim Pharmaceutical (Hanlim Pharm Co. Ltd., Seoul, Korea)) is used in Korea as an adjuvant therapy for lymphadenoma due to breast cancer treatment or to improve symptoms related to venous lymph node failure. Here, we also focused on the pharmacological effect of ETL to prevent the metastatic potential of TNBC through the downregulation of IL1A.

IL1A is one of the prominent inflammatory cytokines and contributes to tumor invasion, angiogenesis, and metastasis by inducing VEGF, TNF-α, HGF, and TGF-β [28,29,30]. Binding of IL1A and its type one receptor (IL-1R1) activates intracellular signaling molecules, including MAPK and NF-κB [13,30,31]. In particular, IL1A mainly stimulates the expression of proinflammatory cytokines, including IL6 and IL8 in colon cancer and melanoma [13,31]. Recently, we reported that the abnormal induction of IL8 triggered the metastatic capacity of TNBC cells [9,32]. In agreement, we also found that the levels of CCL2 and IL8 were increased in TNBC cells by IL1A treatment. Therefore, we demonstrate that IL1A-induced CCL2 and IL8 expression directly or indirectly associated with cancer metastasis in TNBC cells.

Head and neck squamous cell carcinoma (HNSCC) patients with distant metastasis have higher levels of IL1A than those without metastasis [33]. In particular, patients with high IL1A had a significantly lower five-year DMFS than low-IL1A patients [33]. Song et al. reported that the level of IL1A expression was associated with tumor size and lymph node metastasis and significantly correlated with the poor prognosis of patients with human cervical cancer [34]. Consistent with these reports, our results showed that the growth and invasiveness of TNBC cells with IL1A were superior to that of TNBC cells without IL1A. In addition, high IL1A expression in TNBC patients was associated with a poor prognosis. Therefore, we prove that IL1A expression strongly supports in the tumorigenicity of TNBC.

Given that IL1A is produced by tumor cells, it stimulates the proliferation of cervical cancer cells but not normal cervical cells [35]. Amphiregulin, an EGFR ligand, upregulates the level of IL1A expression by activating NF-κB [36]. The promoter activity of IL1A is significantly increased by PU.1 or NF-κB in HER2 overexpressing cells and elevated IL1A production maintains a local inflammatory environment [37]. In addition, a PI3K kinase inhibitor slightly enhanced the level of IL1A expression, possibly due to the inhibitory effects of PI3K/Akt activity on the MAPK kinase pathway [38]. However, our results showed that the basal IL1A level was reduced by the specific MEK1/2 inhibitor UO126 or the specific PI3K inhibitor LY294002. Interestingly, ETL dose-dependently decreased the levels of ERK and AKT phosphorylation in TNBC cells. Therefore, we demonstrate that the transcriptional activity of IL1A in TNBC cells is suppressed by a MEK/ERK- and PI3K/AKT-dependent pathway. Blockage of the MEK/ERK and PI3K/AKT pathway by ETL affected the downregulation of IL1A expression.

## 4. Materials and Methods

### 4.1. Reagents

Entelon^®^ was supplied by Hanlim Pharmaceutical (Hanlim Pharm Co. Ltd., Seoul, Korea). The Human IL1A Quantikine ELISA Kits were purchased from R&D systems (Abingdon, UK). Anti-β-actin was purchased from AbFrontier (Seoul, Korea). Anti-phospho (p) and total (t)-ERK and AKT antibodies were purchased from Cell Signaling Technology (Beverly, MA, USA).

### 4.2. Cell Culture

MDA-MB231 (MDA231), Hs578T, and MDA-MB468 (MDA468) human breast cancer cells were grown in Dulbecco’s modified Eagle’s medium (DMEM, Rockville, MD, USA) supplemented with 10% FBS (Hyclone, Logan, UT, USA), 2 mM glutamine, 100 IU/mL penicillin, and 100 μg/mL streptomycin. HCC1143 and HCC1806 human breast cancer cells and 4T1 mouse breast cancer cells were grown in RPMI1640 media (Life Technologies, Rockville, MD, USA) under silconditions. All cells were maintained in a humidified atmosphere of 95% air and 5% CO_2_ at 37 °C. The cell culture medium was routinely tested for mycoplasma by the EZ-PCR Mycoplasma Test kit (Biological Industries, Beit Haemek, Israel) [5].

### 4.3. Mouse Cytokine Array

To assess the alteration of various cytokines by ETL, we analyzed a mouse cytokine array (R&D Systems, Minneapolis, MN, USA) in culture media of 4T1 mouse breast cancer cells. All procedures were performed following the manufacturer’s protocol.

### 4.4. Analysis of IL1A Clinical Significance

The levels of IL1A mRNA expression was analyzed according to relapse-free survival (RFS) and distant metastasis-free survival (DMFS) in patients with TNBC breast cancer using the Kaplan–Meier (KM) plotter database. DFS (n = 255) and DMFS (n = 43) were analyzed in patients with TNBC. Log-rank *p*-values and hazard ratios (HRs) with 95% confidence intervals were determined on the webpage (https://kmplot.com/analysis/index.php?p=service&cancer=breast, 1 March 2021) [22].

### 4.5. Quantitative Real-Time Polymerase Chain Reaction (qRT-PCR)

Total RNA (1 µg) from each sample was reverse transcribed according to the previously described protocol [5]. Alterations in gene expression were analyzed using SensiMix SYBR kits (Bioline Ltd., London, UK) and the ABI PRISM 7900HT instrument (Applied Biosystems, Foster City, CA, USA). The primer sequences were as follows: mouse IL1A (forward, 5′-ATC AGT ACC TCA CGG CTG CT-3′ and reverse, 5′-TGG GTA TCT CAG GCA TCT CC-3′), human IL1A (forward, 5′-AGT AGC AAC CAA CGG GAA GG-3′ and reverse, 5′-TGG TTG GTC TTC ATC TTG GG-3′), human IL1B (forward, 5′-GCC CTA AAC AGA TGA AGT GCT C-3′ and reverse, 5′-GAA CCA GCA TCT TCC TCA G-3′), human CCL2 (forward, 5′- CAG CCA GAT GCA ATC AAT GC-3′ and reverse, 5′- GCA CTG AGA TCTT CCT ATT GGT GAA-3′), human IL8 (forward, 5′-AGG GTTGCC AGA TGC AAT AC-3′ and reverse, 5′-AAA CCA AGG CAC AGT GGA AC-3′), and GAPDH as the endogenous control (forward, 5′-ATT GTT GCC ATC AAT GAC CC-3′ and reverse, 5′-AGT AGA GGC AGG GAT GT-3′). For data analysis, the raw threshold cycle (C*_T_*) value was normalized to the housekeeping gene for each sample to obtain the ΔC*_T_*. The normalized ΔC*_T_* was calibrated to the control cell samples to calculate the ΔΔC*_T_* [5]*_._*

### 4.6. IL1A ELISA

The protein levels of IL1A were detected according to the manufacturer’s protocol using Human Quantikine ELISA Kits, and then a microtiter plate reader was used to read the plate at a wavelength of 450 nm.

### 4.7. Western Blotting

Whole cell lysates were harvested using PRO-PREP^TM^ Protein Extraction Solution (Intron Biotechnology, Inc., Gyeonggi-do, Korea) and centrifuged (16,100*× g* for 15 min). After quantifying the isolated proteins, the proteins were boiled for 5 min in Laemmli sample buffer and an equal amount (30 μg/lane) of total proteins was loaded to 10% sodium dodecyl sulfate-polyacrylamide gel electrophoresis (SDS-PAGE). Proteins were transferred to polyvinylidene fluoride (PVDF) membranes (Bio-Rad Laboratories, Hercules, CA, USA) and incubated with t-, p-ERK, AKT, and β-actin antibodies in 1% TBS/T buffer (0.01% Tween 20 in TBS) at 4 °C overnight. The blots were washed 3–4 times in TBST and incubated with appropriate secondary antibodies (Santa Cruz Biotechnology, Inc., Santa Cruz, CA, USA) in TBST buffer for 1 h at room temperature (RT). After 1 h, the blots were washed 3–4 times with TBST buffer. The protein expression bands were visualized using ECL prime reagents (Amersham, Buckinghamshire, UK) [5].

### 4.8. Boyden Chamber Assay

As in a previous study [9], 4T1 and MDA-MB231 breast cancer cells (2 *×* 10^5^ cells/well) were resuspended in Matrigel-coated filter inserts (8-μm pore size, Becton-Dickinson, San Diego, CA, USA) and added to the upper compartment of the insert chamber in the presence or absence of 20 ng/mL mIL1A, IL1A, or 30 μg/mL ETL. Fresh culture media (700 μL) was added to the lower compartment of the insert chamber. The chambers were incubated at 37 °C for 16–24 h. After incubation, the upper compartment of the insert chamber was removed using cotton swabs and the bottom filters were fixed and stained (toluidine blue, Sigma-Aldrich, St. Louis, MO, USA). The breast cancer cells that invaded through the Matrigel were located on the underside of the filter. The cells on the underside of the filter were photographed using a CK40 inverted microscope (Olympus, Tokyo, Japan).

### 4.9. Colony-Formation Assays

4T1 and MDA-MB231 TNBC cells (2 × 10^3^ cells/well) were seeded in six-well plates in conditioned culture media. After 24 h, the cells were treated with 20 ng/mL mIL1A, IL1A, or 30 μg/mL ETL in a 37 °C incubator for two weeks. After two weeks, viable colonies were stained with 0.01% crystal violet and observed using a CK40 inverted microscope [5]. 

### 4.10. Cell Cycle Analysis

4T1 TNBC cells were seeded into 60 mm culture dish and incubation for 24 h. Each cell was treated with or without 30 μg/mL ETL for 48 h. After 48 h, cells were trypsinized and washed with phosphate-buffered saline (PBS) twice. Following centrifugation (524× *g* for 5 min at RT), cells were resuspended in 1 mL PBS and fixed in 70% ethanol for 20 min at RT. Fixed cells were centrifuged at 524× *g* 1500 rpm for 5 min and washed twice with PBS. The supernatant was discarded, and cell pellets were resuspended in 1 mL PBS with 100 μg/mL DNase-free RNase A (Biopure) and incubated for 30 min in a 37 °C water bath. Next, 50 μg/mL propidium iodide (Sigma-Aldrich, St. Louis, MO, USA) was added to the cell suspension and analyzed using the FACSCalibur-vantage flow cytometer (Becton Dickinson, San Diego, CA, USA)

### 4.11. Animal Study

Female BALB/c nude mice (Orient Bio, Seoul, Korea) at 6–8 weeks used to establish a nude mouse xenograft model. All animal care and experimental procedures complied with guidelines approved by the Institutional Animal Care and Use Committee (IACUC) of Samsung Medical Center [39].

4T1 TNBC cells (1.8 × 10^7^ cells/120 μL) were resuspended in Matrigel (BD Biosciences, Bedford, MA, USA) and injected directly into the right secondary mammary fat pads. The mice were randomly divided into two groups (n = 5/group), which were treated with water only (vehicle) or ETL in the drinking water (0.2%) starting on the day of inoculation and continuing until the end of the experiment. Once the tumors reached a volume of approximately 50 mm^3^, the tumor size was measured using digital calipers at set time points, and the volume was determined using the formula V = 1/2 × length × (width)^2^. Growth curves were calculated using the average relative tumor volume per group (water or 0.2% ETL-treated) at the set time points [39].

### 4.12. Immunohistochemical Staining

For immunohistochemistry assays, paraffin-embedded primary tumors and lung tissue sections were deparaffinized in xylene, dehydrated in graded alcohol, and hydrated in water. Tissue sections (4 μm) were evaluated by H&E and Ki-67 staining with the appropriate positive and negative controls. TUNEL staining was performed using the ApopTag Peroxidase In Situ Apoptosis Detection kit (Millipore, CA, USA) according to the manufacturer’s instructions. Quantitative staining data for Ki-67 and TUNEL were obtained by counting four fields per slide. The slides were analyzed using a Scanscope XT apparatus (Aperio Technologies, CA, USA) [23].

### 4.13. Statistical Analysis

Data analysis was performed with Microsoft Excel 2016 and GraphPad Prism 8 software (La Jolla, CA, USA). Statistical significance between two groups of data was calculated using the Student’s t-test (two-tailed). Tukey’s multiple comparison test was used for comparisons among multiple groups. The results are presented as the mean ± S.E.M. All stated *p*-values were two-tailed, and the differences were considered significant at *p <* 0.05 [5].

## 5. Conclusions

Conclusively, aberrant IL1A induction elicited tumor growth and invasiveness in the TNBC models. Furthermore, the survival rates of TNBC patients with high IL1A levels were poorer than in those with low IL1A levels. We also proved that ETL significantly reduced the basal IL1A level by inhibiting the MEK/ERK- or PI3K/AKT-signaling pathway in TNBC cells. Therefore, we demonstrate that the IL1A expression level plays a pivotal role in the recurrence of breast cancer and could be a target for TNBC treatment. Thus, we will further strive to change the drug repositioning of ETL by demonstrating its drug efficacy by inhibiting IL1A.

## Figures and Tables

**Figure 1 molecules-26-03644-f001:**
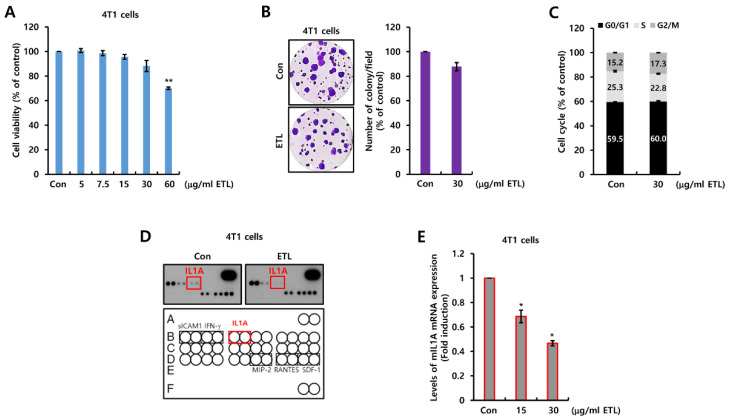
ETL downregulates mouse IL1A suppression in 4T1 TNBC cells. (**A**–**C**) after cell-seeding, 4T1 TNBC cells were treated with the indicated concentrations of ETL for 48 h (**A**,**B**) or two weeks (**C**). (**A**) cell viability was analyzed by the MTT assay. (**B**) the cell cycle was analyzed by flow cytometry. (**C**) cell proliferation was analyzed by the colony-forming assay. (**D**) after 24 h of serum starvation, the cells were treated with or without 30 μg/mL ETL for 24 h. We harvested the conditioned cell culture media and then analyzed mouse cytokines. (**E**) the expression of mIL1A mRNA was analyzed by real-time PCR. The values are presented as the mean ± standard error of three independent experiments. * *p* < 0.05, ** *p <* 0.01 vs. Con. Con; control, ETL; Entelon^®^, mIL1A; mouse IL1A.

**Figure 2 molecules-26-03644-f002:**
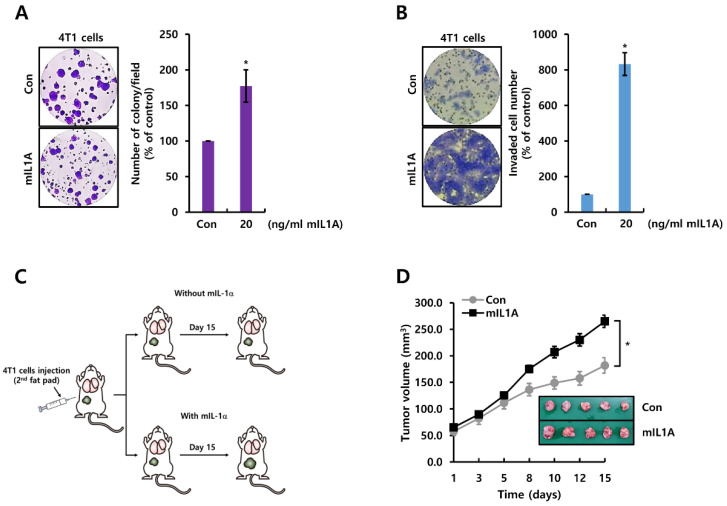
mIL1A augments cell growth and invasiveness in TNBC cells. (**A**,**B**) after cell-seeding, 4T1 TNBC cells were treated with 20 ng/mL mIL1A for two weeks (**A**) or 24 h (**B**). Cell proliferation and invasiveness were analyzed by the colony-forming assay (**A**) and the Boyden chamber assay (**B**), respectively. The upper compartment of the insert chamber was removed using cotton swabs and the bottom filters were fixed and stained for counting invaded cells. (**C**) schematic model of the experimental procedure. (**D**) the size of the tumor in each group (n = 5) was monitored for 15 days. The values are presented as the mean ± standard error of three independent experiments. * *p* < 0.05 vs. Con. Con; control, mIL1A; mouse IL1A.

**Figure 3 molecules-26-03644-f003:**
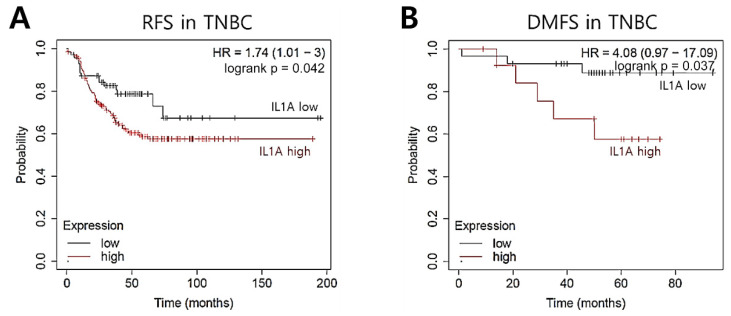
IL1A induction is related to recurrence rates in TNBC patients. The clinical significance of IL1A expression in TNBC patients was analyzed through the KM plotter database. (**A**) RFS, (**B**) DMFS.

**Figure 4 molecules-26-03644-f004:**
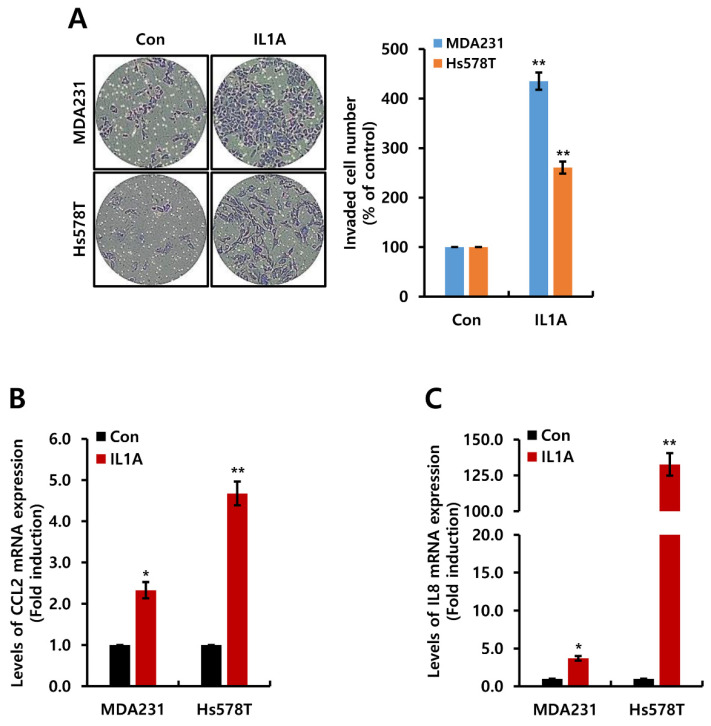
IL1A increased cell invasiveness in TNBC cells. (**A**–**C**) after cell-seeding, TNBC cells were treated with 20 ng/mL IL1A for 24 h. (**A**) cell invasiveness was measured by Boyden chamber assay. The upper compartment of the insert chamber was removed using cotton swabs and the bottom filters were fixed and stained for counting invaded cells. (**B**,**C**) the levels of CCL2 (**B**) and IL8 (**C**) mRNA were measured through real-time PCR. The values are presented as the mean ± standard error of three independent experiments. * *p <* 0.05, ** *p <* 0.01 vs. Con. Con; control.

**Figure 5 molecules-26-03644-f005:**
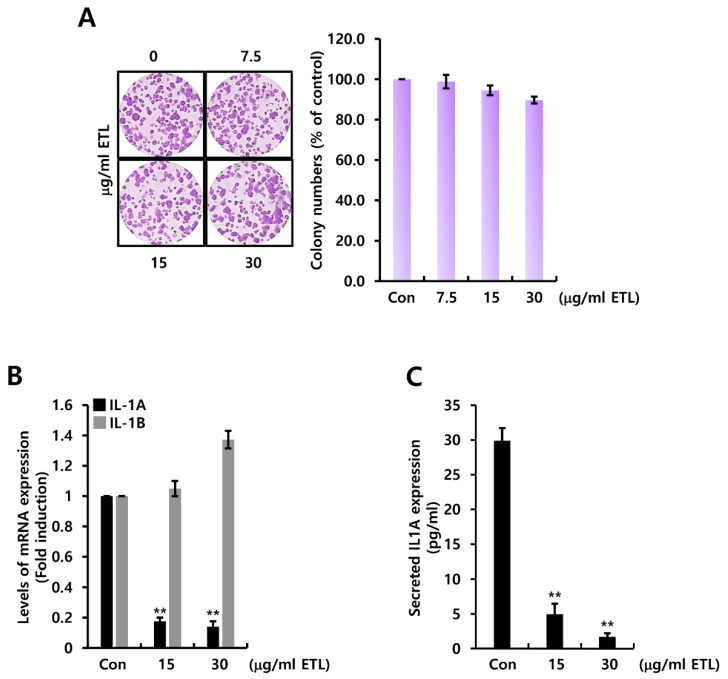
ETL downregulates human IL1A expression in MDA231 TNBC cells. (**A**) after cell-seeding, the indicated concentration of ETL was treated to the cell for two weeks. (**B**) the levels of IL1A and IL1B mRNA were measured by real-time PCR. (**C**) the amount of secreted IL1A protein from conditional culture media was measured by ELISA. The values are presented as the mean ± standard error of three independent experiments. ** *p <* 0.01 vs. Con. Con; control, ETL; Entelon^®^.

**Figure 6 molecules-26-03644-f006:**
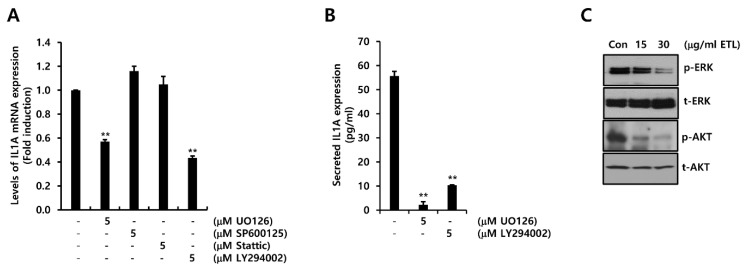
ETL downregulates human IL1A expression by preventing the MEK/ERK and PI3K/AKT signaling pathway in MDA231 TNBC cells. (**A**,**B**) after cell-seeding, 5μM of inhibitor was treated to the cells for 24 h. The levels of IL1A mRNA (**A**) and protein (**B**) were measured through real-time PCR and ELISA, respectively. (**C**) the levels of p- and t-ERK and AKT were measured through Western blotting. The values are presented as the mean ± standard error of three independent experiments. ** *p <* 0.01 vs. Con. Con; control, ETL; Entelon^®^.

**Figure 7 molecules-26-03644-f007:**
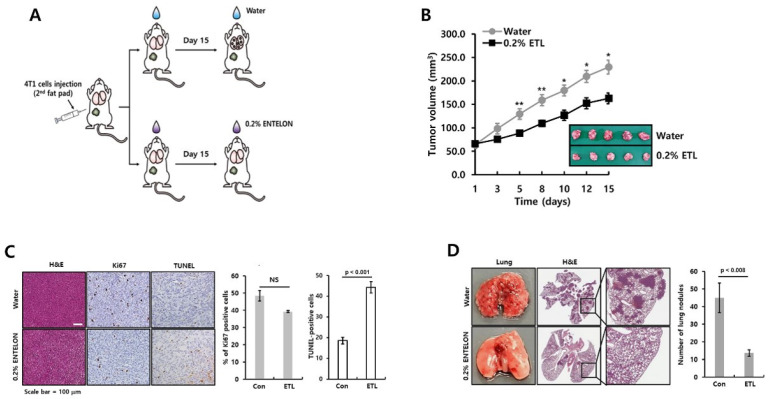
ETL prevents the tumor growth and metastatic capacity of TNBC cells. (**A**) schematic model of the experimental procedure. (**B**) the size of the tumors in each group (n = 5) was measured and analyzed for 15 days. (**C**) the results, confirmed by immunohistochemistry, were scored. Quantitative data were obtained through the calculation of four fields to analyze the experimental results of Ki-67 and TUNEL-positive cells. (**D**) after 15 days, the lung tissues were dissected and analyzed for metastatic events. The number of metastatic nodules was counted in the lung tissue. The values are presented as the mean ± standard error of independent experiments. * *p <* 0.05, ** *p <* 0.01 vs. (−) control. Scale bar = 100 μm.

## Data Availability

Not applicable.

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
