# Peer review of "Entelon® (Vitis vinifera Seed Extract) Prevents Cancer Metastasis via the Downregulation of Interleukin-1 Alpha in Triple-Negative Breast Cancer Cells"

_molecules, 2021, doi:10.3390/molecules26123644_

Round 1

Reviewer 1 Report

In this study by D. You et al, the authors study both mouse and human triple-negative breast cancer (TNBC) cells for effects of grape seed extract (Entelon, or ETL) and, perhaps just as importantly, the associative effects of IL-1alpha (IL1A) on tumor cell growth. They begin by showing that ETL effectively suppresses expression of IL1A in a mouse TNBC cell line and inhibits their growth, a feature further confirmed after injection of the cells in vivo and administering mIL1A, and that ETL can also suppress cell viability; the same cell line was then shown to exhibit increased tumor mass with co-injection of mIL1A. This led the authors to test for a correlation of IL1A expression in TNBC patients and morbidity, which was identified, along with increased invasiveness in two human TNBC cell lines. Data showing decreased growth of the TNBC cells with ETL, along with effects on ERK and AKT, provide a molecular connection as to how ETL might regulate IL1A expression. Finally, they confirm that ETL can work in vivo by administering the molecule in mice injected with TNBC cells and show that metastatic nodules are reduced, indicating that ETL can also suppress metastasis of mammary carcinoma cells. The data are convincing and the authors have nicely shown both in vitro and in vivo results that connect ETL and IL1A, and the potential for ETL as a therapeutic for breast cancer, if not other types of cancer. There are some issues that should be considered by the authors as follows.

1) Lines 43-50, the authors need to describe details of IL1alpha and why that is the primary isotype of IL1 that is the focus of this work, as compared to IL1beta.

2) Line 62, the authors jump from describing IL1 to now indicating their study is on IL1A, so the reasoning behind this should be briefly explained.

3) Line 67, the authors jump into studies of 4T1 TNBC cells as their primary model to test for ETL activities, but they should at least introduce these as a mouse model of TNBC, which would then lead to the use of human cells.

4) Line 71-73, and Fig 1C, the data shown is described under "effect of ETL on the cell cycle”, but this figure is simply showing colony formation, and therefore is part of a growth assay - perhaps a small rearrangement with this figure adjacent to Figure 1A, and described in addition to the MTT results, would be more appropriate.

5) Legend of Figure 1: 1B (line 83-84), the authors should indicate that the graph shows the percentages of cells at different stages of the cell cycle, as determined by PI staining - assuming this was used; this reader could not find the details of this analysis in the M&M section (see below).

6) Lines 96-97, the authors need to introduce the transition from in vitro assays shown in Figs. 1A and 1B, and then the injection of TNBC cells with in vivo studies for Fig 1D, rather than simply stating the schematic is shown.

7) Figure 2B and line 106, the authors should at least mention that they are analyzing and staining cells on the bottom of the filter after cell migration in the classic Boyden chamber (rather than counting cells actually in the bottom chamber), which is visualized with toluidine blue, as shown in the figure.

8) Lines 118-119, the authors should indicate that they are now turning toward use of human TNBC cells for measurements of invasiveness with IL1A, and then simply list the cell lines used, e.g., NDA231 and Hs578T.

9) Line154, the authors should indicate the mechanisms of inhibition by UO126 and LY294002 in Figs. 5A and 5B, and how those correlate with inhibition of MEK and PI3K signaling pathways, which will lead into the results of each being inhibited by ETL in Fig. 5C.

10) Line 211, the authors should indicate that IL-1 acts by inducing "the expression" of the list of growth factors, and again make the distinction between IL1A vs. IL1B (which is important given their results showing effects of ETL is IL1A-specific).

11) Line 228, the phrase "we prove that aberrant IL1A expression plays a pivotal role in the tumorigenicity of TNBC" should be edited - the data does indeed support that increased IL1A expression is associated with increased growth of TNBC cells, poorer prognosis, and increased tumor growth, but stating that it proves a pivotal role is perhaps overstated – suggest toning the statement down a bit to state the results "strongly support" a pivotal role.

12) Materials and Methods: This reader could not find the details of their cell cycle analyses (Fig. 1B).

Minor issues:

1) Line 23, what does IL1A "augment", the actions of the chemokines - this is unclear and should be more specific.

2) Line 221, should IL1 be indicated as "IL1A"?

Author Response

Reviewer 1

1) Lines 43-50, the authors need to describe details of IL1alpha and why that is the primary isotype of IL1 that is the focus of this work, as compared to IL1beta.

à Thank you for your comment. I do not fully understand your comment. As shown in Figure 1D, 1E and 4B, ETL just only suppresses IL1A expression but not IL1B. So, we focused on IL1A expression in TNBC cells. As you know, IL1A and IL1B bind the same receptor and induce identical pro-inflammatory cytokines such as IL6, IL2, and interferons. The key differences in the biology of IL1A and IL1B as follow. Active IL1A acts as an alamin but IL1B does not function as an alamin. In addition, IL1A binds to extracellular domain of IL1 receptor and nuclear function as transcription factor. However, IL1B just only binds extracellular domain of IL1 receptor.

2) Line 62, the authors jump from describing IL1 to now indicating their study is on IL1A, so the reasoning behind this should be briefly explained.

à Thank you for your comment. As mentioned above, IL1A and IL1B have similarities and differences. One of common properties of IL1A and IL1B bind the same receptor and induce identical pro-inflammatory cytokines such as IL6, IL2, and interferons. As shown in Figure 1D, 1E and 4B, ETL just only suppresses IL1A expression but not IL1B. So, we focused on IL1A expression in TNBC cells.

3) Line 67, the authors jump into studies of 4T1 TNBC cells as their primary model to test for ETL activities, but they should at least introduce these as a mouse model of TNBC, which would then lead to the use of human cells.

à That’s good comment. Thank you. As you know, 4T1 cells have been used to study breast cancer metastasis as they can metastasize to the lung, liver, lymph nodes, brain and bone. So, we focused on the pharmacological activity of ETL during breast cancer metastasis process and so, chosen 4T1 cells.

4) Line 71-73, and Fig 1C, the data shown is described under "effect of ETL on the cell cycle”, but this figure is simply showing colony formation, and therefore is part of a growth assay - perhaps a small rearrangement with this figure adjacent to Figure 1A, and described in addition to the MTT results, would be more appropriate.

à Thank you for your comment. As your comment, we switched the order of figures 1B and 1C.

5) Legend of Figure 1: 1B (line 83-84), the authors should indicate that the graph shows the percentages of cells at different stages of the cell cycle, as determined by PI staining - assuming this was used; this reader could not find the details of this analysis in the M&M section (see below).

à That’s good comment. Thank you. It’s my mistake. We added cell cycle method in M&M section.

6) Lines 96-97, the authors need to introduce the transition from in vitro assays shown in Figs. 1A and 1B, and then the injection of TNBC cells with in vivo studies for Fig 1D, rather than simply stating the schematic is shown.

à That’s good point. Thank you. We added your opinion as follows. “Next, we studied the effect of mIL1A on tumorigenesis of TNBC cells in vivo.”

7) Figure 2B and line 106, the authors should at least mention that they are analyzing and staining cells on the bottom of the filter after cell migration in the classic Boyden chamber (rather than counting cells actually in the bottom chamber), which is visualized with toluidine blue, as shown in the figure.

à That’s good point. Thank you. We added your opinion as follows. “the upper compartment of the insert chamber was removed using cotton swabs and the bottom filters were fixed and stained”

8) Lines 118-119, the authors should indicate that they are now turning toward use of human TNBC cells for measurements of invasiveness with IL1A, and then simply list the cell lines used, e.g., MDA231 and Hs578T.

à Thank you for your comment. We revised your opinion and added the list of cell lines.

9) Line154, the authors should indicate the mechanisms of inhibition by UO126 and LY294002 in Figs. 5A and 5B, and how those correlate with inhibition of MEK and PI3K signaling pathways, which will lead into the results of each being inhibited by ETL in Fig. 5C.

à That’s good point. Thank you. Although we have not found transcription factors directly involved in IL1A expression, we believe that MEK/ERK signaling pathway and PI3K/AKT signaling pathway will increase the expression of IL1A through activation of transcription factors. In the meantime, we find that ETL inhibits the activity of ERK and AKT. Based on these results, we believe that the pharmacological activity of ETL inhibits the expression of IL1A through inhibiting MEK/ERK signaling pathway and PI3K/AKT signaling pathway.

10) Line 211, the authors should indicate that IL-1 acts by inducing "the expression" of the list of growth factors, and again make the distinction between IL1A vs. IL1B (which is important given their results showing effects of ETL is IL1A-specific).

à That’s good comment. Thank you. We corrected “IL1A”. As shown reference 29-31, IL1A up-regulates various growth factors including VEGF, TNF-a, HGF, and TGF-b.

11) Line 228, the phrase "we prove that aberrant IL1A expression plays a pivotal role in the tumorigenicity of TNBC" should be edited - the data does indeed support that increased IL1A expression is associated with increased growth of TNBC cells, poorer prognosis, and increased tumor growth, but stating that it proves a pivotal role is perhaps overstated – suggest toning the statement down a bit to state the results "strongly support" a pivotal role.

à Thank you for your comment. We revised your opinion as follows. “we prove that IL1A expression strongly supports in the tumorigenicity of TNBC”

12) Materials and Methods: This reader could not find the details of their cell cycle analyses (Fig. 1B).

à That’s good comment. Thank you. It’s my mistake. We added cell cycle method in M&M section.

Minor issues:

1) Line 23, what does IL1A "augment", the actions of the chemokines - this is unclear and should be more specific.

à Thank you for your comment. We corrected.

2) Line 221, should IL1 be indicated as "IL1A"?

à Yes, we corrected.

Reviewer 2 Report

  1. On line 42 of the first page, the authors should establish that if there is an approved treatment for TNBC, and establish the reference PMID: 32001481 (FDA approves atezolizumab for PD-L1 positive unresectable locally advanced or metastatic triple-negative breast cancer).
  2. In the title of section 2.2, the authors should clarify why they use the expression "Aberrant IL1A expression", since what was done was only the treatment of TNBC cells with mIL1A.
  3. In text of figure 2, the authors should clarify why they use the expression "Aberrant IL1A expression", since what was done was only the treatment of TNBC cells with mIL1A.
  4. In page 4, line 126, the authors should clarify why they use the expression "Aberrant IL1A expression", since what was done was only the treatment of TNBC cells with mIL1A.
  5. Figure 3A has no correlation with B, C and D, so it must be placed independently.
  6. On page 9, lines 228-229 the authors state that there is an aberrant expression; however, they do not present results for this expression. The authors treated TNBC cells with IL1A, which does not demonstrate aberrant expression.

Author Response

Reviewer 2

1.On line 42 of the first page, the authors should establish that if there is an approved treatment for TNBC, and establish the reference PMID: 32001481 (FDA approves atezolizumab for PD-L1 positive unresectable locally advanced or metastatic triple-negative breast cancer).

à Thank you for your comment. We revised your opinion as follows. “Unfortunately, TNBC is a heterogeneous disease without clinically approved therapeutic target drugs against interleukin-1 (IL-1)”

2.In the title of section 2.2, the authors should clarify why they use the expression "Aberrant IL1A expression", since what was done was only the treatment of TNBC cells with mIL1A.

à Thank you for your comment. We revised your opinion as follows. “mIL1A augments cell growth and invasiveness in TNBC cells”

3.In text of figure 2, the authors should clarify why they use the expression "Aberrant IL1A expression", since what was done was only the treatment of TNBC cells with mIL1A.

à Thank you for your comment. We corrected.

4.In page 4, line 126, the authors should clarify why they use the expression "Aberrant IL1A expression", since what was done was only the treatment of TNBC cells with mIL1A.

à Thank you for your comment. We revised your opinion as follows. “TNBC patients with high IL1A were associated with poor prognosis”

5.Figure 3A has no correlation with B, C and D, so it must be placed independently.

à Thank you for your comment. We separated Figure 3 (A, B) and Figure 4 (A, B, C).

6.On page 9, lines 228-229 the authors state that there is an aberrant expression; however, they do not present results for this expression. The authors treated TNBC cells with IL1A, which does not demonstrate aberrant expression.

à Thank you for your comment. We revised your opinion and discarded “aberrant”.
